# Systemic Oxidative Balance Reflects the Liver Disease Progression Status for Primary Biliary Cholangitis (Pbc): The Narcissus Fountain

**DOI:** 10.3390/antiox13040387

**Published:** 2024-03-23

**Authors:** Marcello Dallio, Mario Romeo, Marina Cipullo, Lorenzo Ventriglia, Flavia Scognamiglio, Paolo Vaia, Giorgia Iadanza, Annachiara Coppola, Alessandro Federico

**Affiliations:** Hepatogastroenterology Division, Department of Precision Medicine, University of Campania Luigi Vanvitelli, Piazza Miraglia 2, 80138 Naples, Italy; marcello.dallio@unicampania.it (M.D.); marina.cipullo@studenti.unicampania.it (M.C.); lorenzo.ventriglia@unicampania.it (L.V.); flavia.scognamiglio@unicampania.it (F.S.); paolo.vaia@studenti.unicampania.it (P.V.); giorgia.iadanza@studenti.unicampania.it (G.I.); annachiara.coppola@unicampania.it (A.C.); alessandro.federico@unicampania.it (A.F.)

**Keywords:** primary biliary cholangitis, cholestasis, oxidative stress, hepatic fibrosis

## Abstract

Biological antioxidant potential (BAP) and Reactive Oxygen Metabolites (dROMs) are two tests complementarily assessing systemic oxidative statuses (SOSs) that are never applied in chronic liver disorders (CLDs). We enrolled 41 ursodeoxycholic acid (UDCA)-naïve Primary Biliary Cholangitis (PBC) patients [age: 58.61 ± 11.26 years; females (F): 39], 40 patients with metabolic-dysfunction-associated steatotic livers (age: 54.30 ± 11.21; F: 20), 52 patients with HBV (age: 52.40 ± 8.22; F: 34), 50 patients with (age: 56.44 ± 7.79, F: 29), and 10 controls (age: 52.50 ± 9.64; F: 7). Liver fibrosis and the steatosis severity were determined using transient elastography, and the SOS was balanced using d-ROMs and the BAP test. The gene expressions of superoxide dismutase (SOD1; SOD2) and glutathione peroxidase (GPx1) were evaluated using real-time PCR in advanced fibrosis (AF: F3F4) in patients with PBC. In contrast to other CLDs, in PBC the dROMs and BAP levels were, respectively, directly and inversely correlated with hepatic fibrosis (dROMs, R: 0.883; BAP, R: −0.882) and steatosis (dROMs, R: 0.954; BAP, R: −0931) severity (*p* < 0.0001 all). Patients with PBC also revealed a progressively increasing trend of d-ROMs (F0–F2 vs. F3: *p* = 0.0008; F3 vs. F4: *p* = 0.04) and reduction in BAP levels (F0–F2 vs. F3: *p* = 0.0007; F3 vs. F4 *p* = 0.04) according to the worsening of liver fibrosis. In AF-PBC, the SOD1, SOD2, and GPx1 expressions were significantly downregulated in patients presenting SOS imbalance (SOD1, *p* = 0.02; SOD2, *p* = 0.03; GPx1, *p* = 0.02). SOS disequilibrium represents a leitmotiv in patients with PBC, perfectly reflecting their liver disease progression status.

## 1. Introduction

The disruption of the redox state constitutes a crucial event sustaining the pathogenesis of several chronic diseases in humans [1]. In physiological conditions, the balance between oxidants and antioxidant agents contributes to the preservation of “healthy” redox states in biological systems [2]. The production of reactive oxygen species (ROS) represents major oxidant agents’ source, as well as a natural moment of aerobic life fueling a large part of cellular functions [1,2]. However, excessive generation of these species, by promoting inflammation and toxicity, may result in serious damage to cells and tissues [1]. To face this, protective defenses are naturally predisposed and include non-enzymatic compounds (e.g., vitamin E and glutathione (GSH)) and the action of specific enzymes [e.g., superoxide dismutase (SOD) and glutathione peroxidase (GPx)] [1,2]. When the capacity of these antioxidant systems fails, the level of inactivated ROS rises, and an “unhealthy” redox state takes over, ultimately, determining the oxidative stress [2]. Oxidative stress represents the epiphenomenon of the redox state impairment, which, also considering its close relationship with the inflammation status, has been largely investigated as a predominant pathogenetic background of various chronic liver disorders (CLDs) and autoimmune diseases [1,3,4].

The local oxidative status disequilibrium has amply been demonstrated as a common feature of CLDs, independently from the etiology [viral (HCV and HBV infection) or dysmetabolic (metabolic dysfunction-associated steatotic liver disease (MASLD)], and it plays a crucial role in fueling hepatocellular damage, the progression of fibrosis, and the onset of hepatocellular carcinoma (HCC) [5]. At the same time, the perpetuation of an extra-hepatic chronic low-grade inflammation status, mutually fueling the systemic oxidative stress persistence, constitutes a well-known shared denominator with CLDs of various etiologies (particularly, HBV, HCV, and MASLD) [5].

In the wide scenario of chronic hepatopathies, Primary Biliary Cholangitis (PBC), through simultaneously representing a CLD and an autoimmune disorder, appears as a completely *sui generis* entity. PBC is a widely recognized autoimmune cholestatic chronic liver disease. A predominant proportion (ranging from 90 to 95%) of patients with PBC is represented by females receiving the diagnosis in the 30–65 years interval age [6]. PBC pathogenesis is based on the interaction between immune response and biliary pathways leading to injury and progressing to cholestasis and liver fibrosis. The histological picture is predominantly characterized by destructive cholangitis which contributes to the selective loss of small intrahepatic biliary epithelial cells (BECs). However, the pathogenetic mechanisms driving progressive bile duct loss remain obscure [6].

Recent studies have suggested the involvement of hepatic oxidative stress imbalance in the onset and progression of PBC. In this regard, robust evidence suggests local oxidant stress is a significant feature of early-stage PBC, and it contributes to the worsening of the disease, through apoptosis, premature senescence, and BECs’ loss and destruction [7]. In support of this, the periductal infiltration of myeloperoxidase (MPO)-positive inflammatory cells has been closely associated with cellular senescence in the early stage of PBC. Moreover, recent reports suggested that the protein kinase ataxia telangiectasia-mutated (ATM)/p53/p21WAF1/Cip1 pathway is involved in the induction of cell cycle arrest induced by oxidative stress [8]. Based on this, many studies have subsequently explored the relationship between oxidative stress and ursodeoxycholic acid (UDCA), which represents the first-line therapeutic regimen for patients with PBC [9,10]. Although antioxidant properties have been demonstrated for UDCA [9], recent findings revealed that the protective effect of this medication is limited to early disease stages [10].

Relevantly, the effect of oxidative stress imbalance on PBC could be represented also by an enhanced immune response [4]. Thus, among the frequently used local oxidative stress markers, Malondialdehyde (MDA) constitutes the most important product of lipid peroxidation, whose role in hepatic steatosis, frequently recurring as a histological feature also in PBC, has been largely explored [11,12]. Liver-derived MDA can form an adduct with human serum albumin (HSA) triggering an immune reaction culminating in autoantibodies’ (IgG) production [12]. Consequently, IgG against MDA adducts to HSA and has been proposed as an indirect marker of lipid peroxidation in PBC.

Pathogenetically considering all these recent findings, as well as clinically considering the recurrent extra-hepatic manifestations reported in patients with PBC [13], and considering the overall autoimmune context [3], it appears more legitimate to consider PBC as a “systemic disease”, rather than merely a CLD.

Despite this, extra-hepatic oxidative stress has never been systematically investigated in patients with PBC or correlated with liver disease progression status, making the evaluation of these features an unmet need and a valuable research challenge.

On the other hand, although, as previously mentioned, systemic oxidative stress has been largely highlighted in various CLDs (MASLD, HBV, and HCV) [5], research efforts have predominantly focused on markers of excessive ROS production and have only partially evaluated valid methods for also assessing the potential impairment of the extra-hepatic defense’s oxidative mechanisms in these conditions. Therefore, the complete study of the systemic oxidative balance, as well as the correlation with liver disease progression, still represents a field to be explored for CLDs.

Considering this, investigating the systemic oxidative balance, through going beyond the already demonstrated local redox state disequilibrium, as well as the relationship with liver disease progression status, appeared as a potential novelty in PBC, as well as, in general, in a CLDs scenario.

Biological antioxidant potential (BAP) and Reactive Oxygen Metabolites (dROMs) are two complementary routinely adopted validated tests, widely used to assess the systemic oxidative status (SOS) balance, largely experimented in various chronic human disorder scenarios, and never applied before in clinical CLDs contexts [14,15,16].

Based on all these aims, the present study aimed to evaluate the systemic oxidative balance of UDCA-naïve subjects receiving a first diagnosis of PBC in comparison with healthy individuals and patients affected by CLDs of other etiologies (MASLD, HBV, and HCV), simultaneously investigating the relationship of SOS with the hepatic steatosis and fibrosis severity.

## 2. Materials and Methods

### 2.1. Patients

In this observational study, after signing informed consent, we consecutively enrolled patients receiving a first diagnosis of PBC (n:41), individuals (n:142) receiving a first diagnosis of CLDs of other etiology including ongoing HBV chronic infection (CHB) (n:52), ongoing HCV chronic infection (CHC) (n:50), and MASLD (n:40), as well as a group of healthy controls (n:10).

PBC was diagnosed serologically using the following CPGs-reported-criteria: (1) chronic (at least 6 months) cholestasis [elevated alkaline phosphatase (ALP) or gamma–glutamyl transferase (GGT)] and the presence of anti-mitochondrial antibodies (AMA) (titer > 1:40); (2) chronic cholestasis and specific antinuclear antibodies (ANA) [immunofluorescence (nuclear dots or perinuclear rims) or enzyme-linked immunosorbent assay (ELISA) results (sp100, gp210)] defining the “PBC-AMA negative” (AMA−) entity [13].

HBV and HCV infections were identified when serological and virological evidence of chronic (at least 6 months) ongoing infection occurred: following the current CPGs, the detection of the HCV antibody (HCVAb) with HCV-RNA positivity and HbsAg (HBV surface antigen) positivity with a HBV-DNA viral load > 10^7^ International Unit (IU)/mL defined chronic HCV and HBV infection, respectively [17,18].

Finally, MASLD was diagnosed in the presence of recently Delphi-consensus proposed diagnostic criteria when imaging-detected hepatic steatosis was associated with overweight or obesity, defined as a Body Mass Index (BMI) >25 kg/m^2^, or the presence of specific cardiometabolic risk factors [19].

The enrollment was carried out at the Hepato-Gastroenterology Division of the University of Campania “Luigi Vanvitelli” between January 2020 and January 2024.

The inclusion criteria were age between 18 and 80 years and a novel-received CLDs (PBC, HBV, HCV, or MASLD) diagnosis. Exclusion criteria were alcoholic liver disease (ALD), smoking (past and/or current), the presence of extra-hepatic chronic inflammatory diseases, acute or chronic kidney diseases, rheumatoid arthritis, systemic lupus erythematosus, or other major systemic inflammatory diseases or tumors, ongoing infections, alcohol or drug abuse history, previous neoplasms (including HCC) diagnosis, use of antioxidant based regimens and hepatoprotective drugs in the 12 months preceding the enrollment, decompensated liver cirrhosis (Child-Pugh B and Child-Pugh C) at the moment of the enrollment or in the previous 12 months, and psychological/psychiatric problems that could have invalidated informed consent. Moreover, at enrollment, PBC individuals receiving ongoing or previous (in the last 12 months) UDCA-based treatment were also excluded. Therefore, all the included patients with PBC were UDCA-naïve, and none had undergone a previous (in the previous 12 months) UDCA-based therapeutic regimen. The Alcohol Use Disorders Identification Test (AUDIT-C) questionnaire evaluating alcohol consumption was adopted to exclude ALD patients [20].

After being recruited, to equalize the potential influences of heterogeneous nutritional habits on the general study outcomes, the population was normalized, and all the included patients received a 3-month controlled dietary regimen (detailed reported below), prepared by an expert nutritionist to meet the wishes, tastes, and needs of everyone. To evaluate nutritional habits relative to this period, a professional dietary diary software was used to report an entire week’s food intake, including workdays and weekends, whereas a validated questionnaire was adopted to evaluate physical exercise (Appendix A).

After this 3-month equally prescribed regimen diet, patients were reconvened to receive a specialist visit. On this occasion, (1) the liver fibrosis stage and the entity of hepatic steatosis were analytically determined by performing the two following non-invasive tools (NITs): Liver Stiffness Measurement (LSM) and Controlled Attenuation Parameter (CAP) assessment in CLDs patients [21,22]; (2) the anthropometrical parameters including the determination of BMI by dividing the weight by the square of the height (kg/m^2^), and the clinical data (the complete medical history collection, drug abuse, comorbidities, and the concomitant therapies record, as well as the nutritional and lifestyle-related assessment embracing physical exercise and dietary habits relative to the previous 3 months) were also obtained; (3) a 10 mL venous blood sample (5 mL of serum and 5 mL of plasma) was collected to measure out biochemical parameters (detailed below) and assess SOS. For all patients, the SOS balance was evaluated by using the colorimetric determination of d-ROMs and the BAP test. According to the manufacturers instructions, the simultaneous evidence of d-ROMs > 27.20 mg H_2_O_2_/dL and BAP < 2000 µmol iron/L showed SOS imbalance [23]. Hence, for CLDs patients, dROMs and BAP values were correlated with NITs-assessed liver disease severity (LSM and CAP), and the prevalence of SOS imbalance according to the fibrosis stage was determined.

Finally, the expression statuses of genes notoriously implicated in oxidative stress defense response (SOD1, SOD2, and GPx1) [2] were evaluated for individuals with PBC who were affected with PBC, and compared in patients with AF-PBC presenting and not-presenting SOS imbalance.

The experimental design is reported in Figure 1. The research complies with the Declaration of Helsinki (1975) and has been approved by the ethical committee of the University of Campania Luigi Vanvitelli in Naples (prot. n. 339/2022).

To investigate the oxidative state, in patients with PBC, in comparison with healthy subjects and other etiologies-related CLDs (MASLD, HBV, and HCV), and the correlation between SOS (dROMs and BAP) and liver disease severity, as well as the prevalence of SOS imbalance according to the hepatic fibrosis stage represented the main outcomes of the present study.

### 2.2. Controlled Regiment Diet, Nutritional and Lifestyle-Related Assessment

During the first 3 months of this study, all the enrolled patients were placed on a low-calorie diet (20–25% less than the number of calories needed to maintain their current weight) varying in protein percentage (10 and 20%), fats (20 and 30%, the saturated ones being less than 10%), and carbohydrates (50–60%, sucrose less than 5%). The dietary regimes were prepared by an expert nutritionist to meet the wishes, tastes, and needs of each individual. A dietary diary software was used to report food intake. Dietary habits and food intake were evaluated using a software analysis, Winfood Software 2.0 package (Medimatica s.r.l., Martinsicuro, Italy), of all the enrolled subjects.

Based on the quantity and quality of foods consumed, the program evaluates the energy intake and the percentage of macronutrients and micronutrients in each food. The complete elaboration of intake shows the list of diet components, the ratio among components, the calories, and the subdivisions into breakfast, lunch, and dinner. We recorded the food intake for a complete week, including working days and the weekend. Data were compared with the tables of food consumption and recommended dietary intakes of the Italian National Institute of Nutrition and Food Composition Database in Italy [24]. To assess physical exercise practiced during the 3-month controlled regimen diet, a validated questionnaire composed of a few questions was administered (Appendix A).

### 2.3. Liver Stiffness Measurement and Controlled Attenuation Parameter Assessment

The LSM was obtained using FibroScan^®^ [version 502 (Echosens, Paris, France)] equipped with M and XL probes [21]. When a BMI was >30 or when the ultrasound-measured distance between the skin and liver capsule was >2.5 cm, an XL probe was used. A total of 10 acceptable measurements, defined as successful LSM, through FibroScan^®^ were performed by an expert physician. The criteria used to determine the quality of measurements were the ones proposed by Boursier based on the IQR: “very reliable” (IQR/M ≤ 0:1), “reliable” (0:1 < IQR/M ≤ 0:3 or IQR/M > 0:3 with LS median < 7:1 kilopascal), or “poorly reliable” (IQR/M > 0:3 with LS median ≥ 7:1 kPa) [21,22].

The following LSM cut-off scores were used to identify different liver fibrosis stages in patients with PBC: (a) F0–F2 ≤ 9.8 kPa; (b) F3: 9.9–17.3 kPa; (c) F4 ≥ 17.8 kPa; AF (F3–F4) was defined using LSM values > 9.9 kPa [25]. The following LSM cut-off scores were used to identify the different liver fibrosis stages according to the Metavir score in MASLD: (a) F0–F2 ≤ 9.6 kPa; (b) F3: 9.7–13.5 kPa; (c) F4 ≥ 13.6 kPa; AF (F3–F4) was defined using LSM values > 9.7 kPa [26]. The following LSM cut-off scores were used to identify the different liver fibrosis stages in CHB patients: (a) F0–F2 ≤ 8.1 kPa; (b) F3: 8.2–11 kPa; (c) F4 ≥ 11.1 kPa; AF (F3–F4) was defined using LSM values > 8.2 kPa [27]. The following LSM cut-off scores were used to identify the different liver fibrosis stages in CHC patients: (a) F0–F2 ≤ 9.6 kPa; (b) F3: 9.7–14.6 kPa; (c) F4 ≥ 14.7 kPa; AF (F3–F4) was defined using LSM values > 9.7 kPa [28]. For each CLD etiology, F0–F2 defined “initial–mild” fibrosis, F3–F4 “AF”.

The entity of liver steatosis was analytically assessed at the baseline using CAP [version 502 (Echosens, Paris, France)]. The CAP measures ultrasonic attenuation in the liver at 3.5 MHz using signals acquired by the FibroScan^®^ M and XL probes based on the physical principles described elsewhere [22,26,29]. The CAP was measured only on validated measurements according to the same LSM criteria [22,26,29]. The CAP scores’ cut-offs were S0, no steatosis (0–10% fat; 0–237 dB/m); S1, mild steatosis (11–33% fat; 238–259 dB/m); S2, moderate steatosis (34–66% fat; 260–292 dB/m); and S3, severe steatosis (>67% fat; ≥293 dB/m) [22,26,29].

### 2.4. Biochemical Parameters Assessments

The evaluated biochemical data were aspartate aminotransferase (AST), alanine aminotransferase (ALT), gamma–glutamyl transferase (GGT), alkaline phosphatase (ALP), total bilirubin (TB), albumin, and platelets count (PLT). GGT levels were measured enzymatically using commercially available kits (R&D Systems, Minneapolis, MN, USA), and AST and ALT, using a colorimetric assay kit (Amplite 13801/13803 and Thermo Fisher Scientific EIAGLUC, Waltham, MA, USA).

### 2.5. Systemic Oxidative Stress Assessment

The d-ROMs (colorimetric determination of Reactive Oxygen Metabolites^®^) and the BAP (Biological Antioxidant Potential^®^) tests were used to assess the systemic pro-oxidant and anti-oxidant state, respectively, according to the manufacturer’s instructions (Diacron International, Grosseto, Italy). Blood samples were collected intravenously, and serum was obtained through centrifugation at 2000 rcf for 10 min at RT. Briefly, hydroperoxides (ROOH, primarily) contained in the serum, together with an acidic buffer are converted to alkoxy and peroxyl radicals in the presence of transition metals, which in turn oxidize an alkyl-substituted aromatic amine (*N*,*N*-diethylparaphenylendiamine), showing increased absorbance at 505 nm. The d-ROMs test detects the density of the colored complex photometrically, which is directly proportional to the concentration of hydroperoxides. One unit of the d-ROMs test (U.CARR) corresponds to the amount of hydroperoxide that can be converted by superoxide dismutase to approximately 0.08 mg/dL H_2_O_2_. Reference values defined as the normal range are from 20 to 24 mg/dL H_2_O_2_. The intra- and inter-assay coefficients of variation are 0.3–6.6% and 0.3–5.1%, respectively.

By contrast, the BAP test is settled upon the capacity of the biological sample to reduce iron from the ferric form (Fe^3+^) to the ferrous (Fe^2+^) one. The read-out of the reaction can be assessed photometrically at 505 nm. Data are expressed as µmol of ferric ions reducing antioxidants per liter of samples, with normal reference value >2200 µmol/L.

According to the manufacturer, d-ROMs > 27.20 mg H_2_O_2_/dL and BAP < 2200 µmol iron/L are representative threshold values in building up an SOS impairment.

Both tests were performed in 96-well plates in duplicate (Greiner bio-one) in the end-point method, including a low and high serum as controls as well as an internal calibrator provided by Diacron International. Absorbances were collected with a TECAN INFINITE M PLEX reader (Tecan, Austria). Data are shown as mean ± SD.

### 2.6. RNA EXTRAction and Gene Expression Analysis of SOD1, SOD2, and GPx1 through Real-Time PCR

Evaluating the gene expression of SOD1, SOD2, and GPx1 was performed in AF-patients with PBC. Whole blood underwent density centrifugation with LymphoprepTM, and PBMCs were isolated as previously described [30]. Afterwards, cells were lysed, and RNA was isolated with an RNeasy mini kit (Qiagen, Germany) according to the manufacturer’s instructions. RNA (1 µg) was converted to cDNA using the High-Capacity cDNA Reverse Transcription Kit (Applied Biosystem, Waltham, MA, USA) in a thermocycler (Applied Biosystem) following the optimal conditions.

The expressions of target genes were assessed using real-time PCR (RT-PCR) assays using a Quantstudio 7flex (Applied Biosystems) with PowerUp™ SYBR™ Green Master Mix according to the manufacturer’s instruction.

The primer pairs used for this study (SOD1-5′CGAGCAGAAGGAAAGTAATG3′ and 5′TAGCAGGATAACAGATGAGT3′; SOD2-5′AGTTCAATGGTGGTGGTCATA3′ and 5′CAATCCCCAGCAGTGGAATAA3′; GPx1-5′-TATCGAGAATGTGGCGTCCC-3′ and GPX1- 5′-TCTTGGCGTTCTCCTGAATGC-3′) were designed using Primer3web version 4.1.0 (https://primer3.ut.ee/) and custom-made using Invitrogen. Data were analyzed according to a comparative 2^−ΔΔCt^ method using glyceraldehyde 3-Phosphate Dehydrogenase (GAPDH, 5′AGGGGAGATTCAGTGTGGTG and 5′CGACCACTTTGTCAAGCTCA3′) and β2-microglobulin (β2M, 5′ATGAGTATGCCTGCCGTGTG3′ and 5′CCAAATGCGGCATCTTCAAAC3′) as reference genes. Data are shown as relative gene expression ± SD.

### 2.7. Statistical Analysis

Continuous data were described as mean and standard deviations, while the categorical variables are presented as n (%). The Kolmogorov–Smirnov test for normality was performed to evaluate if the parametric or non-parametric analysis should be applied. Mann–Whitney tests and t-tests were performed for independent groups; Kruskal–Wallis tests or ANOVA tests with posthoc Tukey analyses were performed to compare the continuous variables for non-normal and normal distributions, respectively. The chi-square test was used to evaluate statistically significant frequency distribution differences between the groups. A multilinear regression analysis was adopted to evaluate the relationship (R) between continuous variables.

Statistical significance was defined as *p* < 0.05 in a two-tailed test with a 95% confidence interval (C.I.). GraphPad Prism vs.9.1 was used to perform the analysis.

The sample size was estimated using a chi-square test confronting two independent proportions, singularly predicting a 20% difference in the prevalence of subjects presenting d-ROMs > 27.20 mg H_2_O_2_/dL in each CLD etiology group compared to the healthy one (significance: 0.05, type II error: 0.1; power: 0.9) (STATA14 for MacOS) and resulted in n:10 individuals (for the healthy group) and n:40 patients (for each CLD group, including PBC).

## 3. Results

### 3.1. Study Population: Anthropometric, Biochemical, and Clinical Features

Forty-one patients with PBC, 40 MASLD, 52 CHB, 50 CHC individuals, and 10 healthy controls were enrolled in the present study. The demographic data, anthropometric indexes, biochemical parameters, and NITs for liver disease severity (LSM and CAP) of the study population are reported in Table 1.

The nutritional assessment revealed no difference in dietary habits and food intake in patients with PBC in comparison to CLDs of other aetiologies and healthy ones, both in terms of total daily calorie intake and of quality of the daily intake of macro- and micronutrients during the previous 3-month controlled dietary regimen (Appendix A).

Also, physical exercise in terms of hours/week during this period and the percentage of patients on “active physical exercise” in the last two years were not statistically significantly different in PBC individuals compared to CLDs of other aetiologies and healthy subjects (Appendix A).

### 3.2. Liver Disease Progression Status

The prevalence distribution of AF and severe steatosis (S3) was homogeneous among the study groups and no statistically significant difference when comparing PBC with patients with other CLD aetiologies (MASLD, HBV, and HCV), and MASLD with CHB, CHC with CHB, and MASLD with CHC were evidenced (chi-square comparisons, all *p* > 0.05). In detail, focusing on liver fibrosis, of 41 patients with PBC, LSM revealed 19 (46.34%) presenting initial–mild (F0–F2) fibrosis and 22 (53.65%) AF (F3–F4) (F3: 17 patients; F4: 5 patients). Of 40 individuals with MASLD, initial–mild fibrosis was shown in 17 (42.5%), whereas AF was shown in 23 (F3: 16; F4: 7) (57.5%) patients. Of 52 patients with CHB, LSM revealed 23 (44.23%) presenting initial–mild fibrosis, and 29 (55.77%) with AF (F3: 22; F4: 7). Finally, of 50 individuals with CHC, initial–mild fibrosis was shown in 21 (42%), whereas AF (F3–F4) was shown in 29 (58%) (F3: 16; F4: 13) (57.5%) patients. Focusing on the hepatic steatosis severity, of 41 patients with PBC, 11 (26.82%) were S0–S1, 14 (34.14%) were S2, and 16 (39.04%) were S3. Of 40 patients with MASLD, CAP revealed 10 patients were (25%) S0–S1, 13 (32.5%) were S2, and 17 (42.5%) were S3. Of 52 patients with CHB, 14 (26.92%) were S0–S1, 16 (30.77%) were S2, and 22 (42.31%) were S3. Finally, of 50 individuals with CHC, CAP revealed 12 (24%) were S0–S1, 15 (30%) were S2, and 23 (46%) were S3.

### 3.3. Systemic Oxidative Stress: General Evaluation

Patients with PBC presented higher dROM levels in comparison to controls (*p* < 0.0001), and higher levels were seen in patients with MASLD, CHB, and CHC compared to healthy subjects (healthy: 21.70 mg H_2_O_2_/dL ± 1.15; PBC: 32.64 mg H_2_O_2_/dL ± 5.21; MASLD: 32.91 mg H_2_O_2_/dL ± 8.37; HBV: 30.83 mg H_2_O_2_/dL ± 4.11; HCV: 33.76 mg H_2_O_2_/dL ± 7.88; all *p* < 0.0001) (Figure 2A). On the contrary, lower BAP levels were observed in PBC compared to healthy individuals (*p* < 0.0001), as well as in patients with MASLD, CHB, and CHC in comparison to healthy controls (healthy: 3131 ± 165.2 micromol/L; PBC: 1823 ± 105.7 micromol/L; MASLD: 1697 ± 928.3 micromol/L; CHB: 1961 ± 275.5 micromol/L; CHC: 1516 ± 687.8 micromol/L) (Figure 2B). Relevantly, no statistically significant differences in dROM and BAP levels were reported when patients with PBC, MASLD, CHB, and CHC were compared (Figure 2C,D).

Of 41 patients with PBC, 9 (21.95%) presented an “AMA-negative” (AMA−) pattern. However, no statistically significant differences in BAP and dROM levels were highlighted when comparing patients that were AMA-positive (AMA+) (dROM 30.61 mg H_2_O_2_/dL ± 4.11; BAP: 1795 ± 103.8 micromol/L) and AMA-negative (dROM: 30.97 mg H_2_O_2_/dL ± 3.81; BAP: 1792 ± 101.7 micromol/L) (dROM: *p* = 0.312; BAP: *p* = 0.273).

### 3.4. Systemic Oxidative Stress and Liver Disease Severity

Exploring the relationship between systemic oxidative stress and liver disease progression status, a direct and inverse correlation, respectively, of dROMs (mg H_2_O_2_/dL) (R: 0.883; C.I. 95%: 0.790–0.936; *p* < 0.0001) and BAP (micromol/L) levels [R: −0.882; C.I. 95%: (−0.940)–(−0.780); *p* < 0.0001] with liver fibrosis severity (kPa) was evidenced in patients with PBC (Figure 3A,B). Moreover, the hepatic steatosis entity (db/m) was shown to be directly and inversely correlated with dROMs (mg H_2_O_2_/dL) [R: 0.954; C.I: 95% 0.916–0.975; *p* < 0.0001] and BAP (micromol/L) levels [R: −0.931; C.I. 95%: (−0.963)–(−0.875); *p* < 0.0001] respectively in PBC affected patients (Figure 4A,B). On the contrary, except for a modest positive correlation between LSM (kPa) and dROM levels (mg H_2_O_2_/dL) (R: 0.378; C.I. 95%: 0.075–0.617; *p*: 0.016) in MASLD subjects, no statistically significant correlations between dROM and BAP levels with LSM and CAP were evidenced in other aetiologies for patients with CLDs (Figure 3 and Figure 4).

Relevantly, a statistically significant progressively increasing trend of d-ROMs values (F0–F2 vs. F3: *p* < 0.0008; F3 vs. F4: *p* = 0.04) alongside a reduction in BAP levels (F0–F2 vs. F3: *p* = 0.007; F3 vs. F4 *p* = 0.04) according to the worsening of liver fibrosis were highlighted in PBC, as opposed to individuals with MASLD, CHB, and CHC (Figure 5).

In contrast to other CLDs (MASLD, HBV, and HCV), dROM levels progressively increased also, along with the worsening of hepatic steatosis (S0S1 vs. S2: *p* = 0.0023; S2 vs. S3: *p* = 0.0075), and BAP levels decreased consistently in accordance with the progression from initial–mild to severe steatosis (S0S1 vs. S2: *p* = 0.0023; S2 vs. S3 *p* = 0.0085) in patients with PBC (Figure 6).

In this setting, the multiple linear regression analysis highlighted a positive correlation of dROM levels with the following PBC-related prognostically relevant biochemical variables and scores: GGT (*p* = 0.003), ALP (*p* = 0.0094), AST (*p* = 0.0096), and Albumin (*p* = 0.0431) (Table 2).

Unlike subjects with MASLD, CHB, and CHC, the prevalence of oxidative stress imbalance in PBC was significantly different (*p* < 0.0001) between patients with mild fibrosis and AF fibrosis (Figure 7).

Finally, the real-time PCR analysis revealed a downregulated expression of SOD1, SOD2, and GPx1 genes in patients with AF-PBC presenting SOS imbalance compared with patients with AF-PBC with a balanced systemic oxidative status (SOD1, *p* = 0.02; SOD2, *p* = 0.03; GPx1, *p* = 0.02) (Figure 8).

## 4. Discussion

In recent decades, a solid interplay between oxidative stress and the immune system has progressively emerged configuring the dual driving of the onset and progression of several chronic human disorders [4]. Immune system activities are regularly guaranteed by the correct functioning of relative cells, which is, in turn, physiologically associated with the production of ROS [4]. Oxidative stress represents the epiphenomenon determined by the imbalance between the abnormal formation of ROS and limited antioxidant defenses; alarmingly, in respect to excessive antioxidant defenses, ROS directly interacts with cellular biomolecules, such as DNA, lipids, and proteins, and may result in toxicity, as well as ultimately cause cell death [1,2].

The disruption of local redox state homeostasis has been identified as a crucial contributor to disease worsening in various CLDs, independently from the etiology [5,31].

PBC is a chronic inflammatory autoimmune CLD characterized by progressive cholestasis determining bile duct destruction, fibrosis, and cirrhosis [6]. Despite the great amount of research, it has never been completely clarified whether, at the hepatic level, oxidative stress represents a mere consequence of PBC-related pathogenetic mechanisms or, considering its well-documented pivotal role in disease progression, it also triggers disease onset [7]. On this topic, the strongest evidence derives from studies evaluating the local oxidative status, focusing mainly on lipid peroxidation and indirect antioxidant markers. MDA, one of the most important products of lipid peroxidation, has been shown to able to promote immune response through the formation of adducts with HSA, consequently promoting the formation of circulating antibodies against this complex [12,32]. These specific IgGs have been further used as indirect markers of oxidative stress imbalance. In the PBC context, this evidence strongly supports the crucial role of the oxidative stress status in pathogenesis, given the autoimmune nature of this liver disease [12]. Furthermore, a more sensitive marker is 8-isoprostane, a product of arachidonic acid peroxidation, which can be measured in the blood and urine [33]. On the other side, regarding the antioxidant markers, the most investigated are vitamins A, C, and E, selenium (an essential cofactor of glutathione peroxidase), and GSH [34].

However, all these findings reflect, even indirectly, on the local (i.e., hepatic) oxidative status.

With a translational view of the recurrent extra-hepatic manifestations reported in patients with PBC [13], and, overall, the autoimmune scenario [3], it appears more legitimate to approach PBC as a “systemic disease”, rather than merely localized CLD.

On the other hand, although systemic oxidative stress has been largely highlighted in various CLDs (MASLD, HBV, and HCV) [5], research efforts have only partially evaluated valid methods assessing the potential impairment of the extra-hepatic defense’s oxidative mechanisms in these conditions. Therefore, the complete study of the systemic oxidative balance, as well as the correlation with liver disease progression, represents, in general, an incompletely explored field in the CLDs picture.

These assumptions represented the primum movens to evaluate SOS in the CLD context in our research.

To the best of our knowledge, indeed, in general, for CLD no evidence has been provided on the direct systemic oxidative state assessment, as well as the relationship with disease progression.

Based on all these considerations, we aimed to assess the systemic oxidative balance of UDCA-naïve subjects receiving a first diagnosis of PBC in comparison with healthy individuals and patients affected by CLDs of other etiologies (MASLD, HBV, and HCV), and investigate the relationship with the hepatic steatosis and fibrosis severity. For this purpose, 41 patients receiving a first diagnosis of PBC, 52 CHB, 50 CHC, and 40 MASLD individuals, as well as 10 healthy controls were enrolled.

To avoid the influences of exogenous environmental factors on SOS [35,36,37,38], we voluntarily decided to exclude smokers (currently or previously) and patients with ALD, and given the evidence supporting the potential antioxidant role of UDCA, all the patients with PBC that started UDCA treatment after the collection of serum samples were excluded. None underwent a UDCA-based or systemic antioxidants therapeutic regimen in the previous 12 months. Moreover, to minimize and equalize the well-known impact of lifestyle on the systemic redox state [35,36], before collecting serum samples, all the enrolled patients received a 3-month controlled dietary regimen, prepared by an expert nutritionist. A professional dietary diary software was also used to report an entire week’s food intake, and a specific questionnaire was administered to evaluate the physical exercise (hours/week) practiced during these 3 months. During this observational period, no statistically significant differences, both in terms of dietary habits (quantitative and qualitative estimation of food intake) and physical exercise (hours/week) were reported between healthy patients, patients with PBC, and individuals affected by other CLDs.

Except for variables notoriously featuring specific CLD etiologies (i.e., higher levels of BMI in MASLD and more elevated cholestatic indexes in PBC), the homogeneity of the study population was supported also by the presence of no statistically significant differences emerging for anthropometric, biochemical, and non-invasive tools for liver disease severity assessment when individuals that are healthy and with PBC, MASLD, CHB, and CHC were compared, as well as for the frequency distribution of LSM-assessed fibrosis stages and the CAP-assessed steatosis entity.

Focusing on demographic data, no differences in terms of ethnicity were investigable, since all patients enrolled were Caucasian. Regarding sex distribution, in the PBC group, the prevalence of female patients was significantly higher in comparison to subjects that were healthy, had MASLD, CHB, and CHC. However, considering the epidemiology of PBC, and, in general, of autoimmune disorders [13,39], this has not been considered a selection bias, but rather an accurate portrayal of what occurs in real life.

Therefore, through all these above-mentioned measures conditioning the experimental design, all the selection bias and/or ab-extrinsic elements potentially affecting the SOS evaluation in our study population were avoided.

To assess SOS, BAP, and dROMs, two complementary routinely adopted validated tests, already widely used to determine SOS balances in various chronic human disorders, were performed [14,15,16].

Higher d-ROMs and lower BAP levels in comparison to healthy subjects were shown in PBC, MASLD, CHB, and CHC; interestingly, no statistically significant differences in terms of dROMs and BAP levels were reported between patients with PBC, MASLD, CHB, and CHC. This represents an expected result, in line with the other studies already published, confirming the presence of oxidative stress as a denominator shared by various liver disorders [31]. In patients with PBC, despite existing evidence suggesting peculiarities of the AMA—pattern [40], no differences in dROMs and BAP were revealed between PBC-AMA + and PBC AMA—patients.

However, on one hand, dROM and BAP levels were shown to be higher in all CLD groups; on the other hand, a statistically significant direct and inverse correlation, respectively, of dROMs and BAP with both liver fibrosis and steatosis severity was evidenced exclusively in patients with PBC. Except for a modest positive correlation between LSM and dROM levels in MASLD, indeed, no statistically significant correlations between dROM and BAP levels with LSM and CAP were evidenced in other aetiologies.

Based on these, we further evaluated the dROMs and BAP according to fibrosis and steatosis stages.

Relevantly, in patients with PBC, in contrary to other CLD aetiologies, dROM levels progressively increased along with the worsening of hepatic fibrosis and steatosis, and, consistently, BAP levels accordingly decreased. Despite the same trend being observed also in MASLD, CHB, and CHC, it was revealed as not statistically significant.

Exploring further, some very interesting data emerged, particularly regarding the correlation between liver disease stage progression and OS. We demonstrated a direct and inverse correlation of dROMs and BAP levels with both liver fibrosis and steatosis severity, respectively, highlighting the possible effects of OS on the worsening of the disease stage. The progressively increasing trend of d-ROMs levels and the reduction in BAP levels according to the fibrosis advancement confirmed that the indirect markers of OS change accordingly as the disease worsens.

Focusing on biochemical variables, and potentially investigating the relationship between SOS and disease severity in patients with PBC, the multiple linear regression analysis highlighted an interesting correlation between PBC-prognostically relevant biochemical parameters and dROMs. Particularly, dROMs correlated significantly with albumin, AST, GGT, and ALP.

These results are consistent with the evidence already existing. Indeed, all the variables significantly correlating with OS disequilibrium in our patients with PBC have already been intensively studied and their function is investigated in other diseases. For example, the role of albumin as an antioxidant factor is well known. Albumin is involved in redox processes, as it attracts reactive oxygen and nitrogen species through the free thiol group of Cys34 [41]. Moreover, the correlation with AST is also consistent, since mitochondrial dysfunction is generally a mechanism that supports ROS production and reflects AST mitochondrial isoenzymes’ functioning and release [42]. Furthermore, it is well known that the mitochondrial dysfunction characterizes PBC pathogenesis, as suggested by the fact that its serological hallmark is the “anti-mitochondrial autoantibodies” (AMA) [43]. Regarding “pure” biomarkers of cholestasis, on the one hand, GGT is now regarded as one of the most robust indicators of SOS [44]; on the other hand, ALP is induced by oxidative in vascular and bone cells [45].

Ultimately, we assessed the prevalence of SOS imbalance, defined as the simultaneous presence of d-ROMs > 27.20 mg H_2_O_2_/dL and BAP < 2000 µmol iron/L in every patient: in contrast to MASLD, CHB, and CHC, the prevalence of oxidative stress imbalance was different between patients with mild symptoms and those that were AF-affected in the PBC group.

This result suggests how, in parallel with progressing to the more advanced stages of the disease (F3, F4), the reduced antioxidant defenses crucially contribute to the unbalance of the systemic redox state. This appears consistent with the downregulation of enzymatic antioxidant defenses (including SOD1, SOD2, and GPx functioning). These defense mechanisms are initially intensified and subsequently collapse in the more advanced stages, as already demonstrated for other CLDs [5,31]. In support of this, the RT-PCR-based sub-analysis in AF-affected individuals revealed a statistically significant reduction in SOD1, SOD2, and GPx1 relative gene expression in patients with PBC presenting SOS imbalance.

Our study presents some limitations: firstly, PBC was serologically diagnosed, and liver disease progression status was for all CLD etiologies non-invasively assessed using LSM and CAP. However, following current CPGs, serological criteria represent the way in routine clinical practice to diagnose PBC, reserving liver biopsy to a few limited cases [13]. Regarding the determination of fibrosis and steatosis severity, the transient elastography-based approach (LSM and CAP) constitutes a bulwark, as well as an accurate, commonly, and largely validated tool in the non-invasive staging of CLDs [21,26,29].

Our cohort, despite being representative, was relatively small, and this constitutes another study limitation. However, the exigence of excluding patients in which exogenous factors (including the UDCA administration) would have affected the purity of the SOS assessment by “contaminating” the results significantly impacted on this sample limitation.

Moreover, dROMs and BAP accuracy in the prediction of AF and S3 were voluntarily not assessed: this may represent another limitation of our research. However, various reasons justify this last choice, ranging from the preliminary and observational nature of this research, which explored completely uninvestigated scenarios, to an experimental design that was initially conceived with a completely different aim and, thus, was not adequate (e.g., the sample size was not commensurate) to perform this type of analysis.

Anyway, looking to future perspectives, as previously described, the application of BAP and dROMs as tools to assess SOS is completely new in CLD contexts, and the obtained results open the doors to further investigations, even on larger populations, where these two complementary tests can be applied. In this sense, the evaluation of BAP and dROMs accuracy in stratifying disease progression risk in patients with PBC, as well as predicting UDCA response, would be a crucial and revolutionary milestone in the management of patients with PBC, considering also that dROMs and BAP tests are relatively cheap, and territorial laboratories in Europe are already able to perform them routinely. Alarmingly, there are currently no risk scores used for patients receiving a first diagnosis of PBC, since the currently available prognostic scores have been validated to predict the disease evolution after the treatment starts. The GLOBE and UK-PBC scores, indeed, examine the biochemical parameters 12 and 24 months, respectively, after the introduction of UDCA [46], making the ideation of NITs predicting long-term outcomes in patients with PBC that are UDCA-naïve an unmet need.

## 5. Conclusions

As, in the famous myth, Narcissus dramatically reflects their image into a pond, so the systemic oxidative balance dramatically reflects disease progression in PBC. Preponderantly than other etiologies-based CLDs, individuals with PBC appear to be characterized by a disequilibrium of the oxidative balance whose correlation with liver disease progression remarks its importance as relevant pathogenetic mechanisms that cannot be underestimated.

## Figures and Tables

**Figure 1 antioxidants-13-00387-f001:**
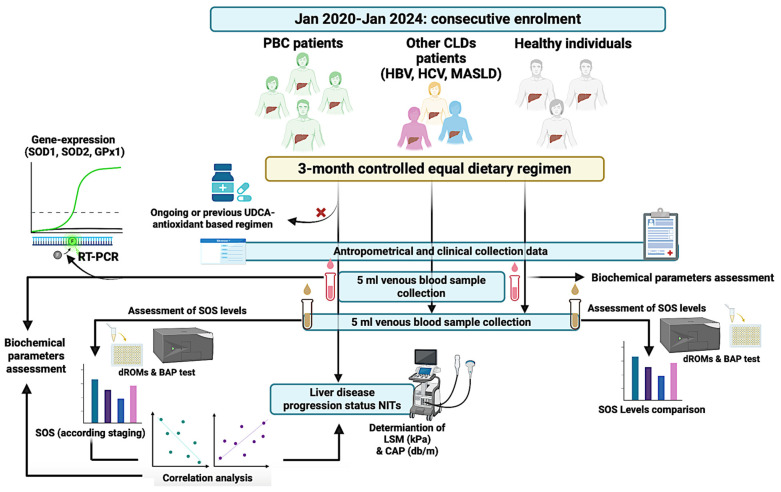
Experimental design. Anthropometrical parameters’ collection enclosed the determination of body mass index (BMI). Clinical evaluation included the complete medical history collection, nutritional assessment, smoking, drug abuse, comorbidities, and the concomitant therapies record. Biochemical variables: aspartate aminotransferase (AST), alanine aminotransferase (ALT), platelets count (PLT), gamma–glutamyl transferase (GGT), alkaline phosphatase (ALP), albumin, total bilirubin (TB). PBC: Primary Biliary Cirrhosis; CLDs: chronic liver disorders; HBV: Hepatitis B virus infection; HCV: Hepatitis C virus infection; MASLD: metabolic dysfunction-associated steatotic liver disease; NITs: non-invasive tools; LSM: liver stiffness measurement; CAP: controlled attenuation parameter; SOS: systemic oxidative stress status; SOD: superoxide dismutase; GPx: glutathione peroxidase; RT-PCR: realtime polymerase chain reaction; UDCA: ursodeoxycholic acid.

**Figure 2 antioxidants-13-00387-f002:**
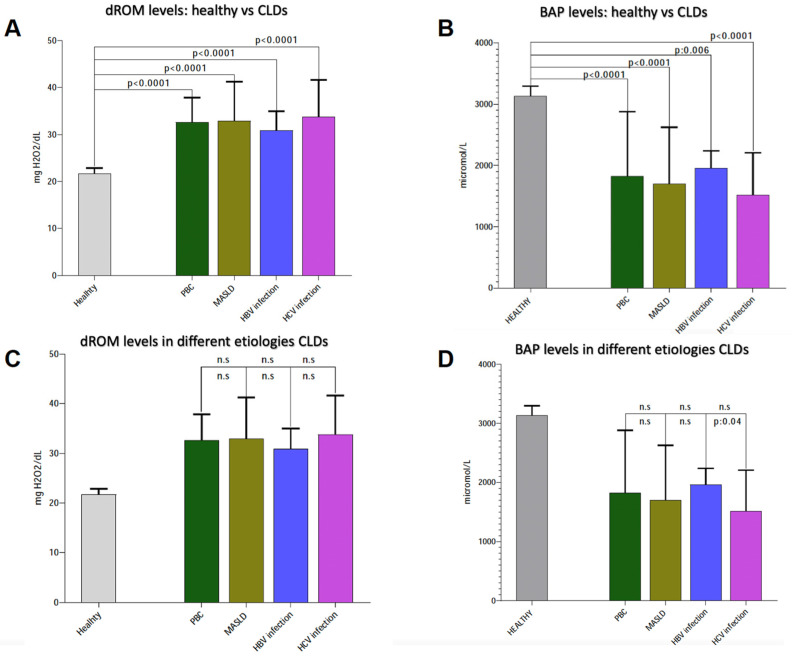
Systemic oxidative stress assessment. (**A**) Comparison of dROM levels between healthy patients and patients with CLDs (PBC, MASLD, HBV, HCV); (**B**) comparison of BAP levels between healthy patients and patients with CLDs (PBC, MASLD, HBV, HCV); (**C**) comparison of dROM levels between different etiologies for patients with CLDs (PBC, MASLD, HBV, HCV); (**D**) comparison of BAP levels between different etiologies for patients with CLDs (PBC, MASLD, HBV, HCV). Biological antioxidant potential, dROMs: Reactive Oxygen Metabolites (dROMs); PBC: Primary Biliary Cholangitis; MASLD: metabolic dysfunction-associated steatotic liver disease; HBV: Hepatitis B infection; HCV: Hepatitis C infection. n.s.: not statistically significant.

**Figure 3 antioxidants-13-00387-f003:**
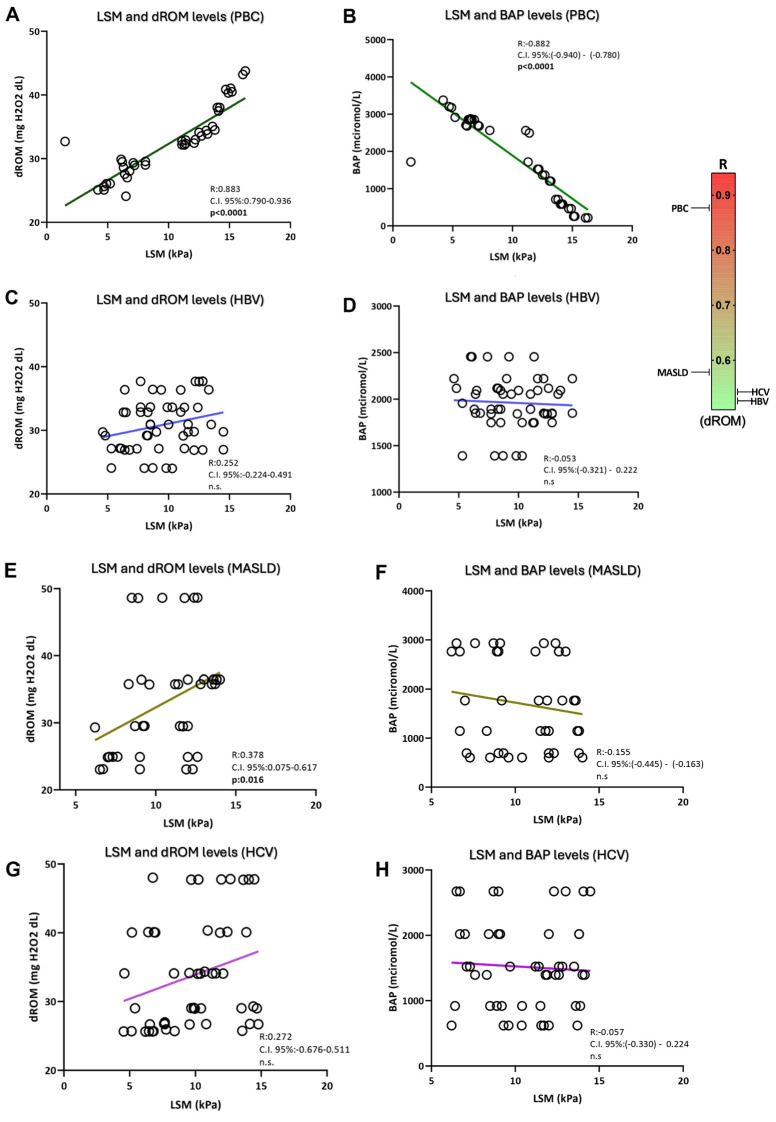
The correlation between systemic oxidative stress and LSM-assessed hepatic fibrosis entity. Linear regression analysis with correlations between LSM and dROM levels (**A**), and LSM and BAP levels (**B**) in PBC; linear regression analysis with correlations between LSM and dROM levels (**C**), and LSM and BAP levels (**D**) in HBV; linear regression analysis with the correlations between LSM and dROM levels (**E**), and LSM and BAP levels (**F**) in MASLD; linear regression analysis with the correlations between LSM and dROM levels (**G**), and LSM and BAP levels (**H**) in HCV. Biological antioxidant potential, dROMs: Reactive Oxygen Metabolites (dROMs); PBC: Primary Biliary Cholangitis; MASLD: metabolic dysfunction-associated steatotic liver disease; HBV: Hepatitis B infection; HCV: Hepatitis C infection. Statistically significant differences (*p* < 0.05) are reported in bold; n.s.: not statistically significant.

**Figure 4 antioxidants-13-00387-f004:**
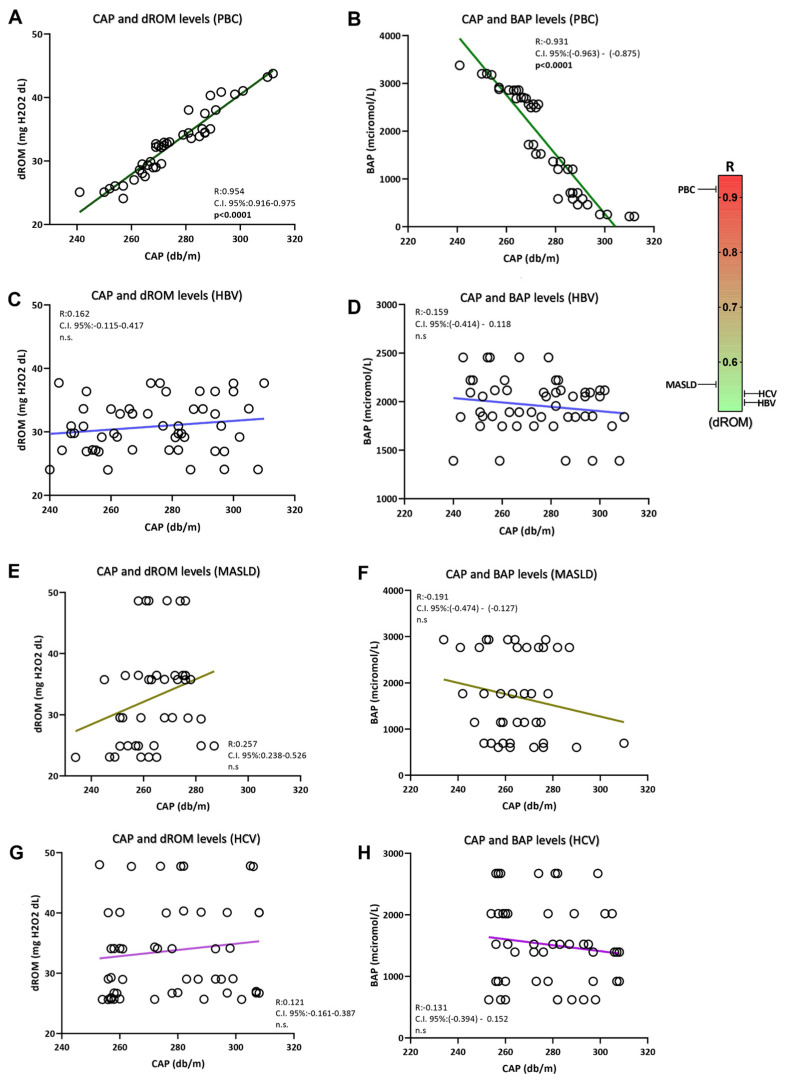
The correlation between systemic oxidative stress and CAP-assessed hepatic steatosis severity. Linear regression analysis with correlations between CAP and dROM levels (**A**), and CAP and BAP levels (**B**) in PBC; linear regression analysis with correlations between CAP and dROM levels (**C**), and CAP and BAP levels (**D**) in HBV; linear regression analysis with correlations between CAP and dROM levels (**E**), and CAP and BAP levels (**F**) in MASLD; linear regression analysis with correlations between CAP and dROM levels (**G**), and CAP and BAP levels (**H**) in HCV. Biological antioxidant potential, dROMs: Reactive Oxygen Metabolites (dROMs); PBC: Primary Biliary Cholangitis; MASLD: metabolic dysfunction-associated steatotic liver disease; HBV: Hepatitis B infection; HCV: Hepatitis C infection. Statistically significant differences (*p* < 0.05) are reported in bold; n.s.: not statistically significant.

**Figure 5 antioxidants-13-00387-f005:**
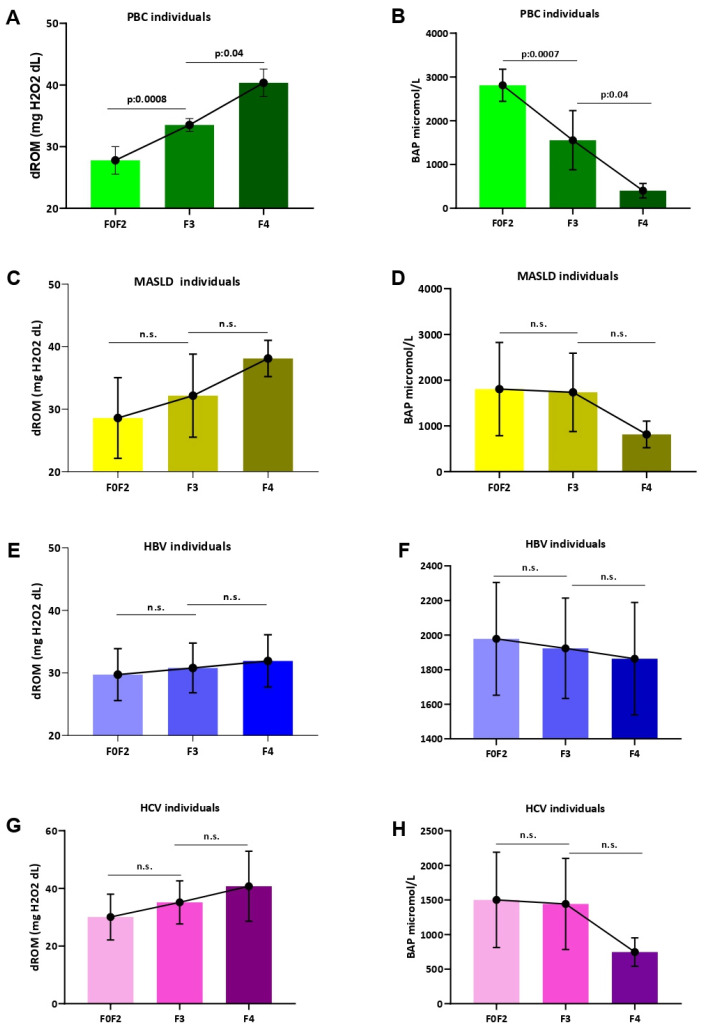
Systemic oxidative stress assessment according to hepatic fibrosis staging. The assessment of dROM values according to LSM-determined hepatic fibrosis stage in patients with PBC (**A**), MASLD (**C**), HBV (**E**), and HCV (**G**). The assessment of BAP levels according to LSM-determined hepatic fibrosis stage in PBC (**B**), MASLD (**D**), HBV (**F**), and HCV (**H**) patients. Biological antioxidant potential, dROMs: Reactive Oxygen Metabolites (dROMs); PBC: Primary Biliary Cholangitis; MASLD: metabolic dysfunction-associated steatotic liver disease; HBV: Hepatitis B infection; HCV: Hepatitis C infection.; Mann–Whitney U test was adopted. Statistically significant differences (*p* < 0.05) are reported in bold; n.s.: not statistically significant.

**Figure 6 antioxidants-13-00387-f006:**
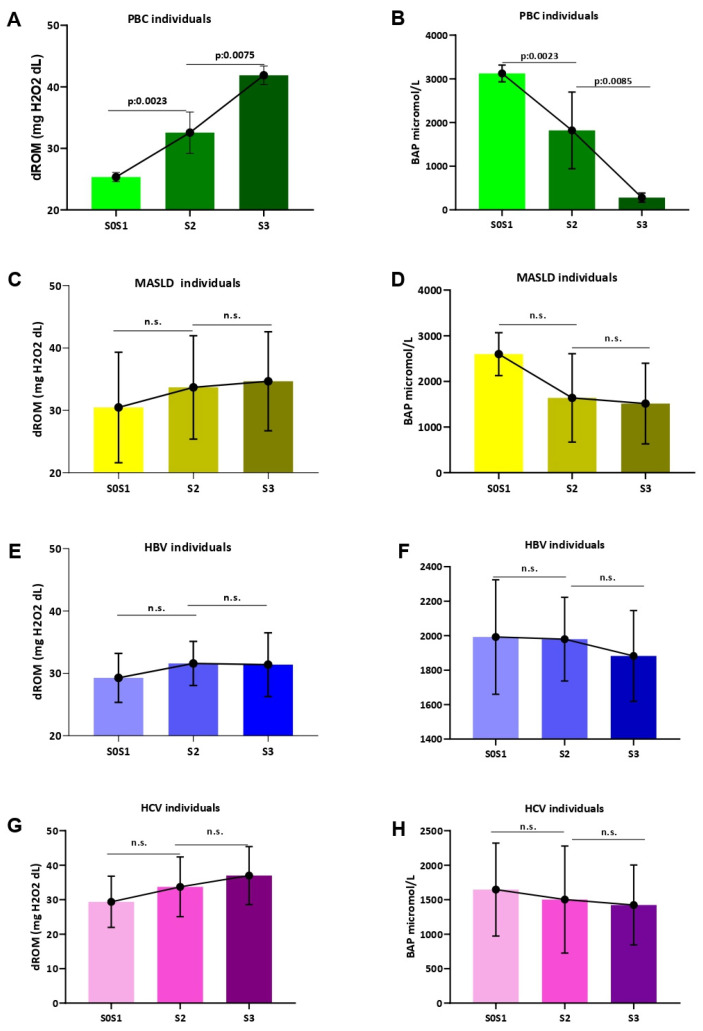
Systemic oxidative stress assessment according to hepatic steatosis severity. The assessment of dROM values according to CAP-determined hepatic steatosis severity in PBC (**A**), MASLD (**C**), HBV (**E**), and HCV (**G**) patients. The assessment of BAP levels according to CAP-determined hepatic steatosis severity in patients with PBC (**B**), MASLD (**D**), HBV (**F**), and HCV (**H**). Biological antioxidant potential, dROMs: Reactive Oxygen Metabolites (dROMs); PBC: Primary Biliary Cholangitis; MASLD: metabolic dysfunction-associated steatotic liver disease; HBV: Hepatitis B infection; HCV: Hepatitis C infection. Mann–Whitney U test was adopted. Statistically significant differences (*p* < 0.05) are reported in bold; n.s.: not statistically significant.

**Figure 7 antioxidants-13-00387-f007:**
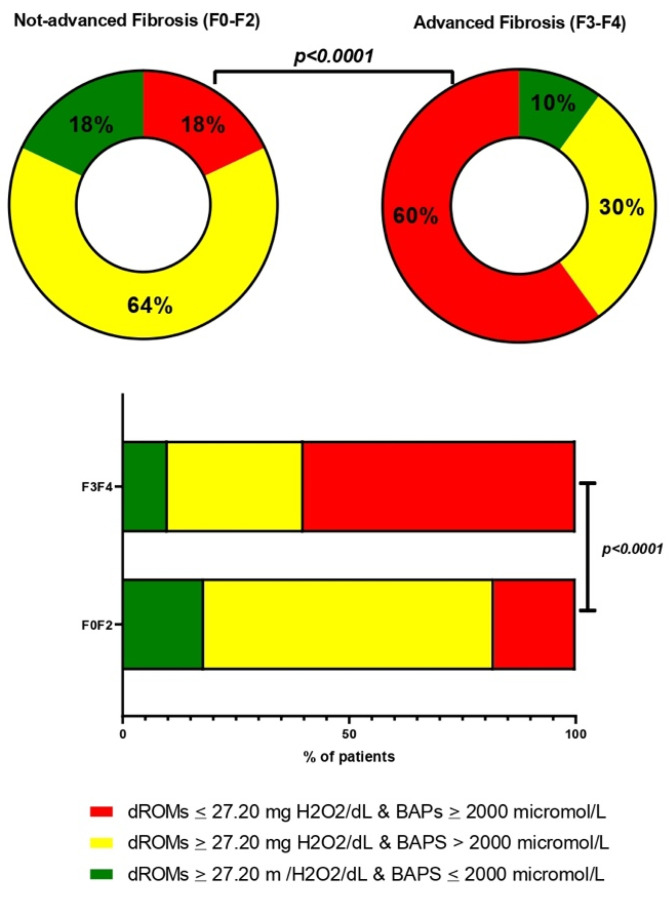
Systemic oxidative stress imbalance prevalence in PBC patients without and with AF. The estimation of systemic oxidative stress imbalance frequency distribution (% of patients) in patients without AF and patients with AF PBC. Biological antioxidant potential, dROMs: Reactive Oxygen Metabolites (dROMs).

**Figure 8 antioxidants-13-00387-f008:**
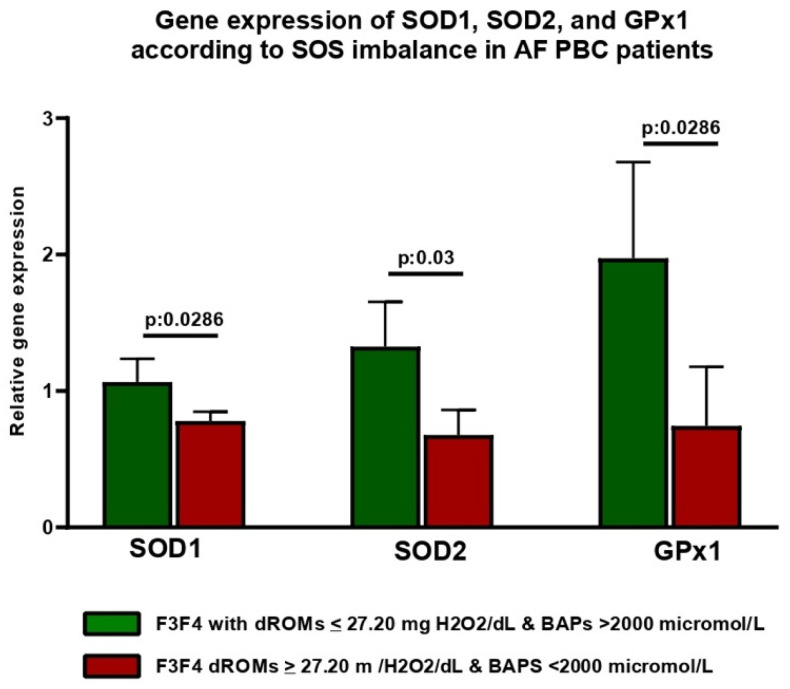
Gene expression of SOD1, SOD2, and GPx1 in patients with AF-PBC. Presenting (red) and not presenting (green) systemic oxidative stress imbalance. Biological antioxidant potential, dROMs: Reactive Oxygen Metabolites (dROMs); PBC: Primary Biliary Cholangitis. Mann–Whitney U test was adopted. Statistically significant differences (*p* < 0.05) are reported in bold.

**Table 1 antioxidants-13-00387-t001:** Demographic, anthropometric, biochemical, and non-invasive tools for the study population.

Demographic Data									
	Healthy (a) (n:10)	PBC (b) (n:41)	MASLD (c) (n:40)	CHB (d) (n:52)	CHC (e) (n:50)	P*/** ab	P*/** bc	P*/** bd	P*/** be
Male (number and %)	3 (30%)	2 (0.048%)	20 (50%)	18 (34.62%)	21 (42%)	**<0.0001**	**<0.0001**	**<0.0001**	**<0.0001**
Female (number and %)	7 (70%)	39 (95.12%)	20 (50%)	34 (65.38%)	29 (58%)				
Age (mean ± standard deviation)	52.50 ± 9.64	58.61 ± 11.26	54.30 ± 11.21	52.40 ± 8.22	56.44 ± 7.79	n.s.	n.s.	n.s.	n.s.
**Anthropometric indexes**									
**Variables (mean ± SD)**	**Healthy (a) (n:10)**	**PBC (b) (n:41)**	**MASLD (c) (n:40)**	**CHB (d) (n:52)**	**CHC (e) (n:50)**	**P* ab**	**P* bc**	**P* bd**	**P* be**
BMI (Kg/m^2^)	29.1 ± 1.21	28.51 ± 3.96	32.3 ± 2.21	27.56 ± 2.61	26.58 ± 3.41	n.s.	**0.03**	n.s.	n.s.
Weight (Kg)	84.20 ± 17.82	73.68 ± 13.08	82.75 ± 14.06	67.58 ± 12.01	72.43 ± 11.05	n.s.	**0.04**	n.s.	n.s.
Height (cm)	162 ± 5.83	158.7 ± 10.71	153.7 ± 12.41	161.3 ± 11.64	159.2 ± 9.62	n.s.	n.s.	n.s.	n.s.
**Biochemical parameters**									
**Variables (mean ± SD)**	**Healthy (a) (n:10)**	**PBC (b) (n:41)**	**MASLD (c) (n:40)**	**CHB (d) (n:52)**	**CHC (e) (n:50)**	**P* ab**	**P* bc**	**P* bd**	**P* be**
AST (IU/L)	17.40 ± 3.74	31.39 ± 13.20	34.38 ± 36.62	42.65 ± 4.46	40.68 ± 5.56	**0.002**	n.s.	**0.02**	**0.03**
ALT (IU/L)	18.40 ± 5.79	32.68 ± 16.55	31.98 ± 19.39	46.88 ± 3.97	46.08 ± 5.36	**0.009**	n.s.	**0.04**	**0.04**
GGT (IU/L)	38 ± 21.01	194.5 ± 69.65	86.93 ± 23.80	87.15 ± 6.11	93.60 ± 10.27	**<0.0001**	**<0.0001**	**<0.0001**	**<0.0001**
ALP (IU/L)	79.50 ± 8.87	192.1 ± 94.99	81.63 ± 44.94	134.9 ± 15.89	105.3 ± 8.81	**<0.0001**	**<0.0001**	**<0.0001**	**<0.0001**
PLT count (10^3^/mm^3^)	198.5 ± 104.4	179.4 ± 18.3	174.8 ± 97.99	175 ± 12.36	170.3 ± 14.25	n.s.	n.s.	n.s.	n.s.
Total Bilirubin (mg/dL)	0.42 ± 0.08	2.06 ± 0.68	1.09 ± 0.79	1.08 ± 0.65	1.14 ± 0.92	**0.006**	**0.002**	**0.002**	**0.003**
Albumin (g/L)	4.39 ± 0.37	3.89 ± 0.42	4.21 ± 0.54	3.99 ± 0.61	3.85 ± 0.22	n.s.	n.s.	n.s.	n.s.
INR	0.93 ± 0.62	1.18 ± 0.29	1.08 ± 0.56	0.96 ± 0.37	1.01 ± 0.54	n.s.	n.s.	n.s.	n.s.
**Non-invasive tools for liver disease severity assessment**									
**Variables (mean ± SD)**	**Healthy (a) (n:10)**	**PBC (b) (n:41)**	**MASLD (c) (n:40)**	**CHB (d) (n:52)**	**CHC (e) (n:50)**	**P* ab**	**P* bc**	**P* bd**	**P* be**
LSM (kPa)	/	10.28 ± 6.02	10.47 ± 6.45	9.94 ± 4.63	9.88 ± 5.05	n.s.	n.s.	n.s.	n.s.
CAP (dB/m)	/	275.1 ± 25.83	274.2 ± 21.65	274.8 ± 29.01	278 ± 28.63	n.s.	n.s.	n.s.	n.s.

BMI: body mass index; AST: aspartate aminotransferase; ALT: alanine aminotransferase; GGT: gamma–glutamyl transferase; ALP: alkaline phosphatase; PLT: platelet count; INR: International Normalized Ratio; LSM: liver stiffness measurement; CAP: controlled attenuation parameter; PBC: Primary Biliary Cholangitis; SD: standard deviation; MASLD: metabolic dysfunction-associated steatotic liver disease; CHB: chronic Hepatitis B infection; CHC: chronic Hepatitis C infection. ** Mann–Whitney U test. * Chi-square test. Statistically significant differences (*p* < 0.05) are reported in bold; n.s.: not statistically significant.

**Table 2 antioxidants-13-00387-t002:** The relationship between dROMs and relevant PBC-related biochemical parameters.

Parameter	t-Statistic	R-Square	C.I. 95%	*p*-Value
GGT (IU/L)	4.92	0.834	0.481–0.977	**0.0003**
ALP (IU/L)	3.04	0.728	0.692–0.954	**0.0094**
AST (IU/L)	3.03	0.548	0.215–0.667	**0.0096**
ALT (IU/L)	2.81	0.536	0.231–0.779	n.s.
Total bilirubin (mg/dL)	2.24	0.318	0.184–0.333	n.s.
PLT count (mm^3^)	0.63	−0.216	−0.421–0.627	n.s.
Albumin (g/L)	1.01	−0.780	−0.231–0.645	**0.0431**

AST: aspartate aminotransferase; ALT: alanine aminotransferase; ALP: alkaline phosphatase; C.I.: confidence interval; GGT: gamma–glutamyl transferase; PLT: platelet count; PBC: Primary Biliary Cholangitis. Multiple linear regression (weighted for BMI, age, and sex); R: correlation matrix. t statistic for each parameter is computed as the parameter value divided by its standard error. Statistically significant differences (*p* < 0.05) are reported in bold; n.s.: not statistically significant.

## Data Availability

Data, analytic methods, and study materials will not be made available to other researchers.

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
