# Peer review of "Systemic Oxidative Balance Reflects the Liver Disease Progression Status for Primary Biliary Cholangitis (Pbc): The Narcissus Fountain"

_antioxidants, 2024, doi:10.3390/antiox13040387_

Round 1

Reviewer 1 Report

In its current form, the article is difficult to read and find their clinically meaningful. According to the Reviewer, it should be thoroughly rebuilt, especially the "Material and methods" and "Results" sections

Abstract:

line 17 lack of abbreviation expansion UDCA- ursodeoxycholic acid

No information about the patients (age and gender of the patients).

Introduction:

1. There exist numerous autoimmune diseases that have an impact on the liver. The age and gender of the patient are important in diagnosis. There is no information on this in the introduction regarding Primary Biliary Cholangitis (PBC)
2. There is no short explanation and connection between patients with PBC, HBV and HCV. The reviewer knows this, but the average reader doesn't, so you should add at least one paragraph. Without this, the selection of research groups is incomprehensible.

Material and methods:

The Reviewer suggests rebuilding this part, because in its current form it is too chaotic and difficult to navigate.
In the Reviewer's opinion, Authors should first start with a description of the patients and the clinical part:
• patient recruitment system
• what were the inclusion and exclusion criteria
• when the material was collected and when FibroScan tests were performed
Laboratory methods should be described at the end of the section.

Figure 1 should be larger because in its current form it is difficult to read

Results:

Question: How did they figure out how many people were in the control group? With so many people in the study groups, it looks like the control group is too small.

Technical note: all tables are prepared inconsistently with the magazine's Microsoft Word template and should be corrected.

Line 491 and 494 are two descriptions for the same drawing?

Discussion:

Lack of "study limitations".

Reviewer 2 Report

The authors are interested in employing the BAP and dROMs tests to assess the SOS in CLDs. They conducted a comparative analysis of the severity of liver fibrosis and steatosis and SOS balance in PBC, MASLD, CHB, CHC groups, and controls. Additionally, they also evaluated gene expression of SOD1, SOD1, and GPx1 in AF-PBC patients. As a result, the authors demonstrated that, in analog to other CLDs, the BAP and dROMs were correlated with hepatic fibrosis and steatosis severity, with an increasing trend of dROMs and reduction of BAP levels. Moreover, specific genes were significantly downregulated in AF-PBC, indicating SOS may represent a leitmotiv in PBC patients. Notably, the authors have acknowledged the limitations of this study. Overall, the study is well-designed, and the manuscript is well-written. However, the figures of this manuscript need to be improved. I have several comments that the authors may consider.

1.     Line 17: please define “UDCA” as “ursodeoxycholic acid”.

2.     Materials and Methods: some contents are redundant. For example, lines 123-125 are repetitive with section 2.3, and lines 133-136 are repetitive with section 2.6. Suggest refining.

3.     Lines 192, 384, 408, 420, 437, 454, and 489. Please move all the figure titles to the figure legends. The titles of Figure 4 and Figure 5 are identical!!

4.     Figures 2: for a better understanding, this reviewer suggests adding a subtitle above each small figure. In addition, Figure 2C should appear behind Figure 2B in the main text. Consider reordering the figure.

5.     Similarly, in Figures 3 and 4, subtitles are recommended for every small figure. Also, I suggest replacing Fig. 3B with 3E and Fig. 4B with 4E.

6.     Figures 5A and 5B: please confirm that p values between F3 and F4 are identical (p=0.04) for bothdROM and BAP; Figures 6A and 6B, please confirm that p values between S0S1 and S2 are identical (p=0.0023) for bothdROM and BAP.

7.     Figure 8: please double-check that the p-values of SOD1 and GPx1 are identical (p=0.0286).

8.     Lines 669-671: this long sentence is hard to follow. Consider dividing it.

9.     Lines 224, 333, 337, and 672-673: please confirm the supplementary materials. The reviewer found only one file and two Tables in the supplementary. Moreover, there is no Supplementary Table S1 in the main text.

10.  Curiously, are there any reasonable explanations for why SOS disequilibrium preponderantly reflects disease progression in PBC compared to other CLDs, including the MASLD?

Round 2

Reviewer 1 Report

Thank you very much for making all the corrections. In its current form, the article is much easier to read and maintains the logical sequence of events of the scientific study.

I accept all the corrections.

Reviewer 2 Report

None

None